# Differential Interactions of Molecular Chaperones and Yeast Prions

**DOI:** 10.3390/jof8020122

**Published:** 2022-01-27

**Authors:** Yury A. Barbitoff, Andrew G. Matveenko, Galina A. Zhouravleva

**Affiliations:** 1Department of Genetics and Biotechnology, St. Petersburg State University, 199034 St. Petersburg, Russia; st035189@student.spbu.ru (Y.A.B.); a.matveenko@spbu.ru (A.G.M.); 2Laboratory of Amyloid Biology, St. Petersburg State University, 199034 St. Petersburg, Russia

**Keywords:** yeast prion, Sis1, chaperone, prion propagation, protein quality control, Hsp104

## Abstract

Baker’s yeast *Saccharomyces cerevisiae* is an important model organism that is applied to study various aspects of eukaryotic cell biology. Prions in yeast are self-perpetuating heritable protein aggregates that can be leveraged to study the interaction between the protein quality control (PQC) machinery and misfolded proteins. More than ten prions have been identified in yeast, of which the most studied ones include [*PSI*^+^], [*URE3*], and [*PIN*^+^]. While all of the major molecular chaperones have been implicated in propagation of yeast prions, many of these chaperones differentially impact propagation of different prions and/or prion variants. In this review, we summarize the current understanding of the life cycle of yeast prions and systematically review the effects of different chaperone proteins on their propagation. Our analysis clearly shows that Hsp40 proteins play a central role in prion propagation by determining the fate of prion seeds and other amyloids. Moreover, direct prion-chaperone interaction seems to be critically important for proper recruitment of all PQC components to the aggregate. Recent results also suggest that the cell asymmetry apparatus, cytoskeleton, and cell signaling all contribute to the complex network of prion interaction with the yeast cell.

## 1. Introduction

Protein misfolding is a common phenomenon which is related to a multitude of human health conditions, some of which are associated with accumulation of protein aggregates [1]. Protein aggregation disorders include such common pathologies as Alzheimer’s disease or Parkinson’s disease caused by aggregation of amyloid beta (Aβ) and α-synuclein, respectively. Some of the pathological protein aggregates can be infectious, causing various diseases of humans and animals, e.g., Creutzfeldt-Jakob disease or chronic wasting disease. Such conditions were termed prion diseases [2] (reviewed in [3]).

All living organisms, from bacteria to humans, have developed a cellular surveillance mechanism to tackle protein misfolding and aggregation. This mechanism, known as the protein quality control (PQC), involves a wide range of proteins. Molecular chaperones are the major components of the cellular PQC system. Chaperones not only actively assist protein folding during or after protein synthesis but also help to either disassemble protein aggregates or target misfolded proteins for sequestration or degradation (reviewed in [4]). Another component of the PQC machinery are the protein-sorting factors that usually help to transport misfolded proteins to sequestration sites ([5,6,7]).

Baker’s yeast *Saccharomyces cerevisiae* is a widely used model organism that is applied to study various aspects of cell biology (reviewed in [8]). Yeast are commonly used as a model for studying PQC processes and components. Some of the yeast proteins may form infectious self-perpetuating protein aggregates (prions) (reviewed in [9]). Such aggregates can be transmitted from cell to cell (e.g., through protein transformation [10]). In contrast to mammalian prions, yeast prions usually represent information units that may drive beneficial phenotypes such as the ability to suppress nonsense mutations [11,12]. Moreover, they were found in many wild yeast strains thus providing an adaptation to certain environments [13]. Clear phenotypic manifestation of yeast prions, as well as the amyloid structure of the prion aggregates (reviewed in [9]), make yeast prions a convenient model to study the relationship between the PQC machinery and amyloids.

Over the last two decades, huge developments have been made in the field of yeast prion biology and the interaction of prions with PQC machinery in yeast. In this review, we would like to summarize the main findings of these and other works and highlight the emerging concepts of amyloid-chaperone interactions.

## 2. Yeast Prions and Their Life Cycle

Over the recent two decades, researchers have discovered more than ten yeast prions (reviewed in [9,14]). The following genetic criteria must hold for a protein-based determinant to be considered a prion: (i) reversible curing, i.e., the ability of a determinant to arise spontaneously after curing; (ii) increased rate of induction upon overproduction of the corresponding protein; and (iii) approximate correspondence between the phenotypic manifestation of the deletion of the corresponding gene and a prion determinant [15].

The most studied of these include [*PSI*^+^], [*URE3*], and [*PIN*^+^] (or [*RNQ*^+^]), the prion forms of the yeast Sup35, Ure2, and Rnq1 proteins, respectively [16,17]. All three of these prions are characterized by the presence of the amyloid aggregates of the corresponding protein [18,19,20,21]. Presence of the [*PSI*^+^] prion causes nonsense suppression, i.e., the ability to read through premature termination codons [11]. [*PSI*^+^] is the amyloid form of the Sup35 protein that normally functions as a translation termination factor [22,23]. Upon prionization, the amount of soluble Sup35 in the cell decreases due to its inclusion into the aggregates, causing translational stop codon readthrough [24]. Another well-studied prion, [*URE3*], is formed by the Ure2 protein which is a nitrogen catabolite repression regulator [16]. Aggregation of Ure2 in the [*URE3*] cells interferes with the normal function of the protein. As a result of it, yeast cells can assimilate poor nitrogen sources in the presence of richer ones [25]. The only known phenotypic manifestation of the [*PIN*^+^] prion is the increased rate of induction of other prions, including [*PSI*^+^] [17,26]. However, the combination of [*PIN*^+^] with another prion, [*SWI*^+^] (the prion form of the chromatin remodelling factor Swi1 [27]) can lead to weak nonsense suppression [28].

Yeast prion proteins typically possess a single prion domain (PrD). PrDs of different protein monomers interact and form the parallel in-register β-sheet structure that corresponds to the core of the fibril [29,30]. PrD of the same prion protein may form different three-dimensional structures (reviewed in [31]). The structural diversity of the amyloid cores drives the phenotypic diversity of yeast prion strains (variants). Such prion variants are characterized by different strength of phenotypic manifestation (e.g., levels of nonsense suppression), amount of soluble protein in the cell, and size of amyloid aggregates [32,33].

Life cycle of all amyloid yeast prions consists of several key processes (Figure 1a), namely (i) assembly of monomers into the amyloid fibrils; (ii) conversion of new monomers into the amyloid conformation and their inclusion into the pre-existing fibril; (iii) fragmentation of fibrils into smaller fragments (prion seeds); and (iv) transmission of seeds into the daughter cell [34]. Recent data suggest an existence of another important step, (v) malpartition or asymmetric segregation of prion seeds, the molecular mechanism of which remains largely undiscovered [35]. The body of published research data indicates that the interaction of yeast prions with the PQC machinery affects all of these stages from nucleation and amyloid assembly [36,37,38] to transmission of prion seeds [39,40]. The main groups of molecular chaperones involved in prion propagation are Hsp104, Hsp70, and Hsp40. Next, we will consider their structure and function in the life cycle of yeast prions in more detail.

## 3. Major Molecular Chaperones Involved in Prion Propagation

### 3.1. Hsp104

Perhaps the most important protein that controls propagation of all known amyloid yeast prions is Hsp104, a chaperone that acts as a disaggregase by rescuing previously misfolded and aggregated proteins [46,47] (reviewed in [4]). Hsp104 is strictly required for maintenance of all known amyloid yeast prions (propagation of non-amyloid prions depends on other chaperone groups—Hsp70 (e.g., for [*SMAUG*^+^], [*GAR*^+^]) or Hsp90 (e.g., for [*ESI*^+^]) (reviewed in [48])). Deletion of *HSP104*, its dominant negative mutation (e.g., Hsp104KT [49,50]) or inhibition with chemical agents such as guanidine hydrochloride (GuHCl) eliminates prions [27,49,51,52] (reviewed in [9]). Hsp104 confers prion propagation by fragmenting the amyloid fibrils of prion proteins into smaller fragments (prion seeds or propagons) that can be then transmitted to daughter cells upon division [47,53,54,55].

Hsp104 is a member of the AAA+ ATPase family and is a homolog of the bacterial ClpA and ClpB proteins which all form ring-shaped hexamers [56]. Hsp104 consists of several key domains (Figure 1b): (i) the N-terminal domain (N-domain or NTD), (ii) first AAA+ nucleotide-binding domain (NBD1), (iii) middle domain (MD), (iv) second nucleotide-binding domain (NBD2), and (v) a small C-terminal domain [57]. NBD1 and NBD2 are involved in the formation of the main channel in the hexamer structure and are responsible for energy-dependent movement during dissolution of client aggregates (NBD1 is mostly responsible for the latter) [57]. During disaggregation of misfolded proteins, Hsp104 employs a ratchet-like mechanism which implies coordinate up- and downward movement of protomers due to ATP hydrolysis [58]. The N-terminal domain is not directly involved in ratchet movements; however, NTDs might act as a regulatory lid that blocks the hexamer channel. Such a configuration has been observed for the bacterial Hsp104 homolog, ClpB, in a free (not substrate-bound) state [59]. Upon substrate binding, hydrophobic surfaces of NTDs interact with a substrate inside the main pore of ClpB. Such a positioning of NTDs suggests that they might play a role in transfer of substrate from other chaperones to Hsp104/ClpB. In concordance with this model, NTD of Hsp104 is involved in substrate binding and interaction with Hsp70 [60]. Moreover, several studies have shown that the N-domain of Hsp104 directly interacts with Sup35NM aggregates [54,61].

Despite the critical importance of Hsp104 for prion propagation, excess of Hsp104 protein can also cure [*PSI*^+^] [49] and destabilize some strains of [*URE3*] [62]. Modest overproduction of Hsp104 also destabilizes [*PSI*^+^] [63]. It is widely acknowledged that such overexpression-mediated curing occurs via a different mechanism. For example, deletion of the N-terminal domain of Hsp104 abrogates overexpression curing of [*PSI*^+^] while not affecting its propagation [42,54]. Several different models of overexpression-based curing of yeast prions by Hsp104 have been proposed. The first model implies that Hsp104 overexpression prevents prion seeds from entering the bud during division of the yeast cell [35,39]. The second model implies that Hsp104 overproduction leads to lack of proper complex formation with other chaperones and results in inefficient binding of the disaggregase to fibrils [54]. The third model proposes that increased levels of *HSP104* expression enhance the shearing capacity of Hsp104 which results in active cleavage of monomers from ends of existing fibrils [64]. Irrespective of the actual mechanism, the critical role of N-domain in prion curing by excess Hsp104 emphasizes the importance of substrate binding for this process. Notably, Hsp104-mediated curing of yeast prions also occurs under normal levels of Hsp104, as prion variants formed in strains bearing a T160M mutation (Figure 1b) in the N-domain of Hsp104 are rapidly lost upon introduction of the wild-type *HSP104* allele [65].

### 3.2. The Hsp70/Hsp40 Chaperone System

Among other chaperone groups, Hsp70 and Hsp40 proteins also play a major role in yeast prion propagation. Hsp70 and Hsp40 proteins assist Hsp104 in disassembly of protein aggregates [46]. Hsp70 represents the main group of molecular chaperones in eukaryotic cells and has a wide range of functions (reviewed in [66]). A typical structure of an Hsp70 chaperone consists of the nucleotide-binding domain (NDB) which is responsible for ATP hydrolysis and the substrate-binding domain (SBD) which usually comprises a lid region [4,67,68]. During binding to the substrate, hydrolysis of an ATP molecule by NBD of Hsp70 promotes a conformational change that leads to the capture of a substrate molecule by SBD and lid domains [69]. In yeast, Hsp70 proteins are encoded by multiple genes of the *SSA* (Stress Seventy sub-family A) and *SSB* (Stress Seventy sub-family B) families [70,71]. The *SSA* subfamily includes four main members (*SSA1-4*) which serve various functions; at least one of the four *SSA* genes must be present for viability. The SBD of Ssa proteins can be divided into two regions, SBDα (including the lid) and SBDβ. The C-terminus of Ssa1 contains an intrinsically disordered region ([43] Figure 1b). Ssb proteins, on the other hand, are non-essential and are associated with the ribosome [72]. The role of Hsp70 proteins in propagation of yeast prions has been identified by Newnam et al. [73]. In this study, overexpression of *SSA1* antagonized prion curing by excess Hsp104. Later, Jung et al. confirmed the importance of *SSA1* for prion propagation by isolation and subsequent characterization of a dominant *SSA1-21* mutation that destabilized [*PSI*^+^] [74,75,76]. In contrast to the Ssa proteins, overproduction of Ssb1 enhances prion curing by excess Hsp104 [77] and destabilizes some prion variants [78,79]. Finally, another member of the Hsp70 family, Ssz1, has been recently implicated in prion propagation [80]. Ssz1 is the component of the ribosome-associated chaperone complex (RAC) which counteracts prion induction, possibly due to its main function in nascent chain folding [80].

The Hsp40 proteins are a large family of cochaperones with more than 20 members in the yeast cells [81]. Hsp40 are commonly called J-proteins as they contain an evolutionary conserved J-domain named after the bacterial Hsp40 DnaJ, the first member of the family [67]. Many J-proteins (including the major cytosolic ones) have a specific domain structure that consists of three key domains: the N-proximal J-domain, G/F-rich domain, and the C-proximal substrate-binding domain (CTD) [67,81]. The main function of the J-protein is to stimulate the ATPase activity of Hsp70 and determine the specificity of their function [82]. This stimulation is performed by the J-domain which interacts with Hsp70 via a critically important HPD motif [83]. On the other hand, the CTD in the J-proteins is responsible for interaction with client proteins (reviewed in [68]). Out of all J-proteins, the most important in terms of their role in prion propagation are double β-barrel J-proteins located in the cytosol—Sis1 and Ydj1. Ydj1 is the most abundant of the yeast J-proteins followed by Sis1 [84].

J-proteins are broadly divided into three main classes that differ in their domain composition [67,81]. Class I J-proteins (such as Ydj1) contain Zn^2+^-binding domain (Figure 1b) while class II proteins (such as Sis1) do not. Some members of class II J-protein family (e.g., Sis1) also have a G/M rich domain preceding the C-terminal substrate binding domain (CTD). This G/M-domain is important for the function of class II J-proteins as shall be detailed below. Both class I and class II J-proteins usually act as dimers and possess a dimerization domain (DD) located at the very C-terminus.

The role of Hsp40 in the propagation of yeast prions was demonstrated by Sondheimer et al. in 2001 [85] and further corroborated by multiple studies [44,86,87,88,89] Sis1 is necessary for maintenance of most yeast prions [85,86,90] while Ydj1 is not [91]. Such a requirement for Sis1 activity represents the general preference for class II J-proteins for disaggregation which is seen across species (e.g., [92]). Data suggests that the main function performed by Sis1 and other J-proteins in prion propagation is the delivery of substrates for Hsp104 [87] (reviewed in [55]). At the same time, J-proteins are involved in the prion curing mediated by *HSP104* overexpression [44,93,94].

### 3.3. Other Proteins

Components of the Hsp90 chaperone system, including Hsp90 itself and its cochaperones, have also been implicated in prion propagation. Inhibition of Hsp90 activity with radicicol or by deletion of the *HSP82* and *HSC82* genes interferes with prion curing by excess Hsp104, as does the deletion of the *STI1* and, to a lesser extent, *CPR7* genes, encoding cochaperones of Hsp90 or Hsp70 [95,96]. The latter deletions rescue [*PSI*^+^] from curing by *SSA1-21*, suggesting the same mechanism of [*PSI*^+^] loss in both cases [97]. Deletions of the *STI1*, *HSP82*, and *HSC82* genes have also been shown to negatively impact prion curing by normal levels of Hsp104 [65].

In addition to the main chaperone systems described above, additional pathways are important for prion propagation and are directly or indirectly involved in regulation of PQC activity. These include intracellular protein sorting factors such as Cur1, Btn2, and Hsp42 [98,99,100], inositol polyphosphate (IP) biosynthesis pathway [101], ubiquitin-proteasomal system [102], and the cell asymmetry apparatus (e.g., Sir2) [40]. The involvement of these systems in prion propagation additionally highlights the complexity of cell-prion interactions.

## 4. Differential Effects of Chaperones and Protein-Sorting Factors on Yeast Prion Propagation

As discussed in the previous section, multiple groups of molecular chaperones and protein-sorting factors are involved in propagation and/or curing of yeast prions. However, it is also important to note that many of these proteins do not only play a role in prion propagation but are affecting propagation of different prions or prion variants in the opposite directions [62,91,100] In this section, we will provide additional details regarding such differential effects (summary data are shown in Table 1).

### 4.1. Hsp104 and Its Variants

As already mentioned above, while Hsp104 is required for propagation of all amyloid yeast prions, its overproduction only cures [*PSI*^+^] and destabilizes some variants of [*URE3*] [49,62,63]. The different requirements of prions and prion variants for Hsp104 activity were further demonstrated in the experiments involving Hsp104-ClpB chimeric proteins. While some such chimeric proteins are capable of propagating several yeast prions, presence of the M-domain of Hsp104 is strictly required for supporting prion propagation [87,115]. Interestingly, the chimera that contains only the M-domain of Hsp104 in place of the M-domain of ClpB propagates [*URE3*], but not other prions [115]. At the same time, the 444B variant containing ClpB NBD2 instead of Hsp104 NBD2 supports propagation of most prions and prion variants [105,115], but can only cure some prion variants upon overproduction [116].

### 4.2. Hsp70

While all groups of chaperones tend to differ in their effects on prions, Hsp70 and Hsp40 proteins have arguably the most pronounced differential effects on prion propagation (reviewed in [91]). Differential impact of Hsp70 expression on different prions and prion variants has been identified in multiple studies [74,78,104]. Specifically, *SSA1* has been shown to cure [*URE3*] but not [*PSI*^+^] [78,104]. Moreover, *SSA1* was shown to strengthen the [*PSI*^+^] suppressor phenotype while preventing its curing by excess Hsp104 [73]. Several works demonstrated that *SSA2* does not cure [*URE3*] [104,117]. However, in contrast to the aforementioned results, we found that *SSA2*, but not *SSA1* overexpression slightly weakens [*URE3-1*] variant [62], thus, sensitivity to either Ssa1 or Ssa2 might depend on the prion variant or strain background. These findings are especially interesting given the high level of sequence identity and functional redundancy among the Ssa proteins [71,118]. In addition to the opposite effects of *SSA1*/*SSA2* overexpression, the *SSA1-21* mutation that destabilizes [*PSI*^+^] does not affect propagation of [*URE3*] and most [*PIN*^+^] variants [63,105]. Moreover, we could not confirm a dominant negative effect of *SSA1-21* overexpression in some of the [*PSI*^+^] strains (Figure 2a). Indeed, the significant destabilization of [*PSI*^+^] by the *SSA1-21* mutation was detectable only when it remained as the only *SSA1* allele; additional deletion of *SSA2* led to complete loss of [*PSI*^+^] [74]. However, such strains could still propagate [*URE3*], but introduction of an analogous mutation into *SSA2* (*SSA2-21*) destabilized both [*PSI*^+^] and [*URE3*] [63]. Given that Ssa2 is a major cytosolic Ssa protein, the differential effects on [*PSI*^+^] and [*URE3*] can be explained by different tolerance to imbalance between mutant and native Ssa proteins.

### 4.3. Sis1 and Its Domain Activity

Hsp40 proteins are also exerting differential effects on different yeast prions and prion variants. As mentioned previously, Sis1 and Ydj1 are the two major cytosolic J-proteins in yeast. Sis1 is required for maintenance of all amyloid yeast prions which have been tested so far [86,90]. While Sis1 depletion by decreasing its expression levels eliminates [*PSI*^+^], [*URE3*], and [*PIN*^+^], their sensitivity to such depletion is markedly different. Specifically, [*URE3*] is the most sensitive to low levels of Sis1 and is lost after 6–8 generations upon its downregulation, [*PIN*^+^] is maintained for as many as 10–15 generations, while [*PSI*^+^] is cured only after 20–40 generations at low levels of *SIS1* expression [86]. At the same time, depletion of Sis1 from the cytosol due to increased expression of chaperone-sorting factors or by direct relocalization into the nucleus may enhance phenotypic manifestation of [*PSI*^+^] while curing [*URE3*] [100]. Expression of different *SIS1* alleles also influences prion variant competition [119], highlighting the differential requirements for Sis1 activity in different prion variants (Table 1).

**Figure 2 jof-08-00122-f002:**
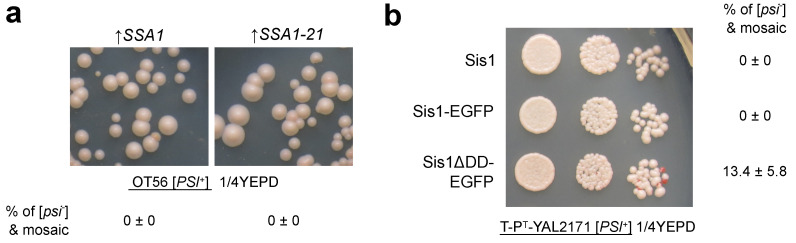
Novel effects of molecular chaperones on propagation of yeast prions. (**a**) Overexpression of *SSA1-21* does not destabilize [*PSI*^+^] in the OT56 strain [73]. Shown are representative colonies of the OT56 strain bearing strong [*PSI*^+^] variant transformed with either pTEF-SSA1 [120] or pTEF-SSA1-21 (obtained using site-directed mutagenesis from pTEF-SSA1) plasmids for overproduction of wild-type Ssa1 or Ssa1-21, respectively, regulated by *TEF1* promoter (**b**) Deletion of Sis1 dimerization domain may destabilize [*PSI*^+^]. Shown are representative 10-fold serial dilutions plated onto 1/4 YEPD medium [121] of the T-P^T^-YAL2171 strain ([*PSI*^+^] derivative of YAL2171 [99]) expressing either full-length Sis1 or a variant with the deletion of dimerization domain (Sis1ΔDD) as the sole source of endogenous Sis1. In the top row, untagged *SIS1* is expressed from its endogenous promoter. In the middle and bottom rows, the *SIS1* gene is fused with *EGFP* on centromeric plasmid and expressed from under the constitutive *ADH1* promoter. The red color of the colonies on the 1/4 YEPD medium indicates the loss of [*PSI*^+^]. On both (**a**,**b**), percentages of [*psi*^−^] and mosaic colonies are estimated from no less than three independent replicates with at least 30 colonies counted for each replicate. Mean and standard deviation are shown in each case.

In addition to the general differences in their dependence on Sis1, yeast prions and their variants also have different requirements for Sis1 domains. For example, [*URE3*] and most [*PIN*^+^] variants are eliminated upon deletion of the Sis1 G/F domain [85,89,105]. At the same time, [*PSI*^+^] can be maintained upon deletion of the G/F region [44,86,88]. Likewise, deletion of the entire substrate-binding domain of Sis1 cures some [*PSI*^+^] variants (similar effects can be seen upon deletion of the dimerization domain (Figure 2b)), [*URE3*], and [*PIN*^+^], while maintaining stronger [*PSI*^+^] [44,88,89]. Remarkably, changes in the Sis1 domain structure have slightly different impact on Hsp104-mediated curing of [*PSI*^+^]: deletion of the CTD of Sis1 or the dimerization domain located at the very end of the protein substantially decreases levels of prion curing by elevated *HSP104* expression [44]. Moreover, deletion of the dimerization domain stabilizes [*PSI*^+^] in strains carrying the *SSA1-21* variant [44], confirming again that these two ways of [*PSI*^+^] elimination follow the same mechanism. A similar effect is observed for the K199A mutation in the CTD of Sis1 (Figure 1b) that decreases its affinity to the substrate and is involved in Hsp70 interaction [83,122]. These findings emphasize the important role of substrate binding for prion curing by excess Hsp104 and for propagation of some, but not all, prion variants. The importance of binding to aggregates is further indicated by the critical role of Sis1 CTD in suppressing prion-related toxicity in yeast cells which is observed upon substitution of wild-type *SIS1* allele with variant containing only J and G/F domains [123].

### 4.4. Other J-Proteins

While Sis1 plays a crucial role in propagation of most yeast prions, another major cytosolic J-protein, Ydj1, also shows differential impact on their maintenance. It has been shown that increased expression of *YDJ1* cures [*URE3*] [51]. However, similar perturbation can lead to both enhancement and destabilization of [*PSI*^+^] depending on the prion variant and strain background [100,109]. Overexpression of the J-domain of Ydj1 is sufficient to cure [*URE3*] while expression of mutant variants of the protein incapable of interacting with Hsp70 does not [86,107]. Similar effects of J-domain overexpression were observed for other J-proteins (e.g., Jjj1 [86]). Furthermore, substitution of the Sis1 CTD with the same domain of Ydj1 cures both [*PSI*^+^] and [*URE3*] [89], supporting the critical role of substrate binding for proper regulation of prion propagation by Hsp40. Hypotheses regarding the mechanistic basis for these effects will be discussed later.

In addition to Sis1 and Ydj1, other J-proteins are also capable of influencing yeast prion propagation. For example, the yeast homolog of mammalian auxilin, Swa2, has been identified as one of the factors strictly required for propagation of [*URE3-1*] variant [110,124]. In addition to Swa2, another member of the class I J-protein family, Apj1, regulates prion propagation and curing by excess Hsp104 [94,106]. Similarly to Sis1 and Ydj1, different prion variants have different requirements for Apj1 activity, with only the strong [*PSI*^+^] variant requiring Apj1 for curing by overexpression of *HSP104*. A characteristic feature of Apj1 is the intrinsically disordered structure of its QS domain (homologous to the G/F domains of Sis1 and other J-proteins) which is rich in glycine, glutamine, and serine. In contrast to Sis1, the CTD of Apj1 is not required for prion curing, while the QS region is necessary for this process [125]. While a similar role of the G/F region was observed for Sis1 [44], it is noteworthy that the Apj1-161 variant completely lacking CTD can promote prion curing by overexpression of *HSP104* [125]. This observation suggests that Apj1 might directly interact with fibrils via its disordered QS region. Alternatively, the QS domain might inhibit interaction between Apj1 and Hsp70 (as shown for the G/F domain of other J-proteins [126]), leading to less Ssa sequestered to aggregates thus favoring aggregate malpartition instead of fragmentation. Yet another Hsp40 that differentially affects yeast prions is the Zuo1 protein that is a part of the aforementioned RAC complex. Zuo1 is not required for propagation of [*URE3*], [*PIN*^+^], or [*PSI*^+^] [86,94,110]; however, its deletion enhances both rates of [*PSI*^+^] induction [80,112] and enhances [*PSI*^+^] destabilization by mild heat shock [40]. Zuo1 interacts with other components of the RAC complex such as Ssb1 and Ssz1; the effects of *zuo1*Δ on prion propagation are heavily dependent on the presence of other RAC components [40,80].

### 4.5. Protein-Sorting Factors

Apart from members of the Hsp40/Hsp70 families, protein sorting factors also affect different yeast prions in opposite directions. For example, protein-sorting factors Cur1 and Btn2 were identified as proteins that cure [*URE3*] [98] and have further been shown to eliminate the artificial [*NRP1C*^+^] prion [99]. Later, Wickner et al. identified a crucial role for Cur1 and Btn2 in eliminating most newly formed [*URE3*] variants under its normal levels [113]. Moreover, deletion of both *CUR1* and *BTN2* genes increased mitotic stability of [*URE3*] variants formed by mutant Ure2 protein [127]. These findings suggest that Cur1 and Btn2 represent an anti-prion defense system in yeast [128,129]. However, overexpression of the *CUR1* gene has been shown to enhance phenotypic manifestation of both strong and weak [*PSI*^+^] prion [100], although this effect seems to be variant-specific as overproduction of Cur1 is capable of curing at least some [*PSI*^+^] strains [109,113]. Activity of both Cur1 and Btn2 proteins depends on their interaction with Hsp40-Sis1, and relocalization of Sis1 into the nucleus is the likely mechanistic basis for the effects of Cur1 on prions [99,100]. However, it was shown that Cur1 and Btn2 act independently of each other at least on [*URE3*] [113]. An alternative hypothesis for Btn2-mediated [*URE3*] curing postulates that it might directly interact with prion aggregates thus affecting their sorting [98,113]. Both Cur1 and Btn2 proteins are unstable and rapidly degraded by the proteasome [99]. Deletion of the N-terminal degron of Cur1 or inhibition of proteasome activity by deletion of the proteasomal genes increases effects of Cur1 and Btn2 [62,100,102].

Finally, one more protein, Hsp42, deserves attention as showing noticeable differential effects on prions. Hsp42 acts as an aggregation-promoting factor or aggregase. Hsp42 is responsible for transport of misfolded substrates and aggregated proteins to insoluble protein deposits (IPOD) [6,7]. Hsp42 efficiently eliminates yeast [*URE3*] prion, as well as an artificial [*NRP1C*^+^] prion [99,100,113]. Likewise, excess Hsp42 was shown to antagonize [*PSI*^+^] [114]. However, we showed that at least some [*PSI*^+^] variants are enhanced rather than cured by *HSP42* overexpression [62,100]. Similarly to Cur1 and Btn2, Hsp42 cooperates with Hsp40-Sis1 to serve its functions [7,99]. Hence, it may be hypothesized that the effect of Hsp42 on yeast prions is also indirectly mediated by changes in Sis1 localization and/or activity.

## 5. Mechanistic Basis for the Differential Effects of Chaperones on Yeast Prions

In the previous section, we extensively described the observed differential effects of PQC components on yeast prion propagation. Next, we will discuss possible explanations for the observed roles chaperones play in the life cycle of different prions and prion variants. Sis1 and Hsp104 emerge as the key regulators of both fibril fragmentation and prion seed malpartition. As thus, any possible mechanism of determining the fate of amyloid aggregates in the cell must be dependent primarily on Sis1 and Hsp104 activities.

### 5.1. The Central Role of Sis1 in Protein Aggregation and Disaggregation

Sis1 appears to be one of the master regulators of aggregation and disaggregation in yeast. Being one of the essential cytosolic J-proteins, Sis1 assists Hsp70 during protein folding. This function of Sis1 may prevent amyloid formation, as shown in several studies [36,37,38,130]. During heat stress, Sis1 cooperates with the stress-inducible aggregases and chaperone-sorting factors to promote protein trafficking to sequestration sites such as JUNQ (JUxtaNuclear Quality control compartment) or INQ (IntraNuclear Quality control compartment) [7,99]. Sis1 is also important for the import of misfolded proteins into the nucleus for degradation on the intranuclear proteasomes [131]. In addition to its role in protein sorting, Sis1 participates in the dissolution of stress-inducible membraneless organelles such as stress granules, as does Ydj1 [132]. Furthermore, Sis1 alleviates [*PSI*^+^] and [*PIN*^+^] prion toxicity as well as prion-dependent toxicity of polyQ aggregates [133,134,135,136]. As shown in a recent study by Klaips et al., Sis1 potentiates stress response to polyQ aggregates by binding to protein oligomers and directing them towards more soluble gel-like state [137]. An increased sensitivity of some strong [*PSI*^+^] strains, as compared to weak [*PSI*^+^] and [*psi*^−^] cells, to some mutations in Sis1 leads to another type of prion-dependent lethality. In this case substrate binding seems to also play a major role, as the JGF variant of Sis1 as the only source of Sis1 is unable to support viability of strains with several strong [*PSI*^+^] variants [44,123], however, this lethality could not be observed in a similar study by another group [88].

Taken together, these findings suggest that Sis1 regulates formation and dissociation of both solid and liquid assemblies inside the cell. We can assume that Sis1 should possess the ability to discriminate between misfolded proteins that should be targeted for disaggregation and those to be transported for sequestration or degradation. As discussed earlier in this review, interaction with substrate plays a critical role in many Sis1 functions; hence, we may hypothesize that the mode of Sis1 interaction with substrate might play a role in determining the fate of the misfolded protein or a protein aggregate.

Indeed, affinity of the Hsp40 interaction with the substrate influences disaggregation activity of other chaperones [138]. For class II J-proteins such as Sis1, we may hypothesize that high affinity to the substrate may interfere with the crucial interaction between CTD of Hsp40 and Hsp70 that alleviates the inhibitory effect of the G/F domain [126]. The inhibitory activity of the G/F domain may also explain the requirement of this region in Sis1 (and the QS region of Apj1) for prion curing. Recently, we developed a method for quantitative assessment of chaperone binding to amyloid fibrils of yeast prion proteins in vitro [139]. Using this method we showed that Sis1 binds Sup35NM fibrils with much higher affinity compared to Ydj1, while the binding of Sis1 to amyloids of Rnq1 is substantially weaker. Importantly, we also showed that the deletion of the dimerization domain of Sis1 decreases its affinity to the Sup35NM fibrils [139]. Given that Sis1 is required for malpartition of [*PSI*^+^], the unique high affinity of Sis1 to amyloid aggregates of Sup35 may play a role in malpartition. However, to explain our interpretation of this observation we shall further consider additional details regarding the Hsp104 function.

### 5.2. Modes of Hsp104 Function and the Mechanisms of Malpartition

As discussed previously, Hsp104 does not play a role in the main cycle of protein folding and acts mostly as a disaggregase that catalyzes dissociation of pre-existing aggregates. Similarly to Sis1, Hsp104 plays an important role in the dissociation of stress-induced membraneless condensates such as stress granules [140]; however, Hsp104 does not play a role in disaggregation of liquid gels formed upon pH stress [141]. Hsp104 also plays a role in polyQ toxicity, but actively suppresses it only in the mutant form bearing an additional A503V substitution in the M-domain [133]. Hsp104 also cooperates with Hsp40 proteins and Hsp42 in regulating age-related protein deposits in yeast [142,143]. Such age-related protein aggregates are asymmetrically inherited; similar asymmetric inheritance is observed for damaged and misfolded proteins [144]. Asymmetric inheritance of protein aggregates requires Hsp104 interaction with the actin microfilaments [145]. It has been proposed that actin refolding required for asymmetric segregation of aggregates is regulated by the Sir2 protein [146].

The involvement of Sir2 in controlling prion seed malpartition, as shown by Howie et al. [40], allows us to hypothesize that prion seeds might be tethered to the actin cytoskeleton by Hsp104 during malpartition. Such a tethering should occur when the Hsp104 directly interacts with the prion aggregates with its N-domain. Indeed, such a direct (not involving any cochaperones) interaction has been demonstrated previously [54,61], and should occur at a higher rate when the chaperone balance is shifted towards Hsp104 upon its overproduction or mild heat stress [147].

If malpartition requires direct interaction of Hsp104 with prion aggregates, why are the J-proteins (and their substrate binding) so important for this process? Two alternative models may be proposed to answer this question. The first model implies that tight binding of Sis1 or Apj1 to prion seeds may prevent efficient recruitment of Hsp70-Ssa to aggregates, which in turn increases the possibility of direct interaction between Hsp104 NTD and a propagon. An alternative model predicts that Sis1 bound tightly to an amyloid fibril is not capable of functionally interacting with Hsp70 but can instead interact with the NTD of Hsp104 to facilitate its binding to an aggregate in a non-productive manner. Both models seem plausbile and are not mutually exclusive (Figure 3). However, further experiments are required to test the validity of these assumptions.

Both tight interaction of Hsp40 with protein aggregates and their direct binding by the NTD of Hsp104 may represent two parts of the same mechanism by which a eukaryotic cell identifies terminally misfolded or damaged proteins to prevent their transmission and inheritance. If true, leveraging these mechanisms could offer new therapeutic prospects for fighting human amyloidosis and other diseases caused by protein misfolding and aggregation.

## 6. Conclusions

Yeast prions provide an important and useful framework for disentangling the complex interactions between the protein quality control machinery and misfolded proteins and amyloids. Over the last two decades, most groups of molecular chaperones have been implicated in prion propagation; moreover, the majority of these chaperones differentially affect different yeast prions and/or prion variants. Such differential effects are most pronounced for members of the Hsp40 family (Sis1, Ydj1, and others) which play a central role in specifying the functions of Hsp70 and Hsp104 [68].

Taken together, all of the data on differential effects of chaperones on prions allow us to expand the previously proposed model of prion-chaperone interactions (Figure 4). For the [*PSI*^+^] prion, two antagonistic chaperone-mediated processes are active: fibril fragmentation and prion seed malpartition. The former process is aided by either Sis1 or Ydj1, while the latter strictly requires Sis1 and Apj1 and is tightly controlled by the cell asymmetry apparatus (mainly Sir2). At the same time, Ssb1 and Zuo1 play a negative role in [*PSI*^+^] propagation by preventing prion conversion of new monomers or fibril fragmentation. For [*URE3*] or [*PIN*^+^], on the other hand, prion seed malpartition acts at a much lower rate or is completely inactive. At the same time, fragmentation of [*URE3*] or [*PIN*^+^] fibrils heavily depends on Sis1 and Swa2, while Ydj1 not only does not support the process but actively interferes with it. For all prions, protein-sorting factors Cur1, Btn2, and Hsp42 regulate the availability of Sis1.

While the differential effects of chaperones on yeast prions are well characterized, little is known about the mechanistic basis of these effects. For example, it is yet unclear why [*PSI*^+^] is substantially more sensitive to curing by excess Hsp104 but shows lower requirements for Hsp40 activity. Multiple lines of evidence suggest that the ability of chaperones to bind directly to amyloid aggregates plays a central role in their impact on prion propagation. Moreover, recent findings emphasize a possible link between cytoskeleton and prion inheritance in yeast. It can be hypothesized that the differences in affinity towards amyloid aggregates of different prions represents a universal mechanism by which the yeast cell discriminates between different types of misfolded proteins and protein aggregates to determine their fate in the cell. However, the nature and meaning of the observed differences certainly require further investigation.

## Figures and Tables

**Figure 1 jof-08-00122-f001:**
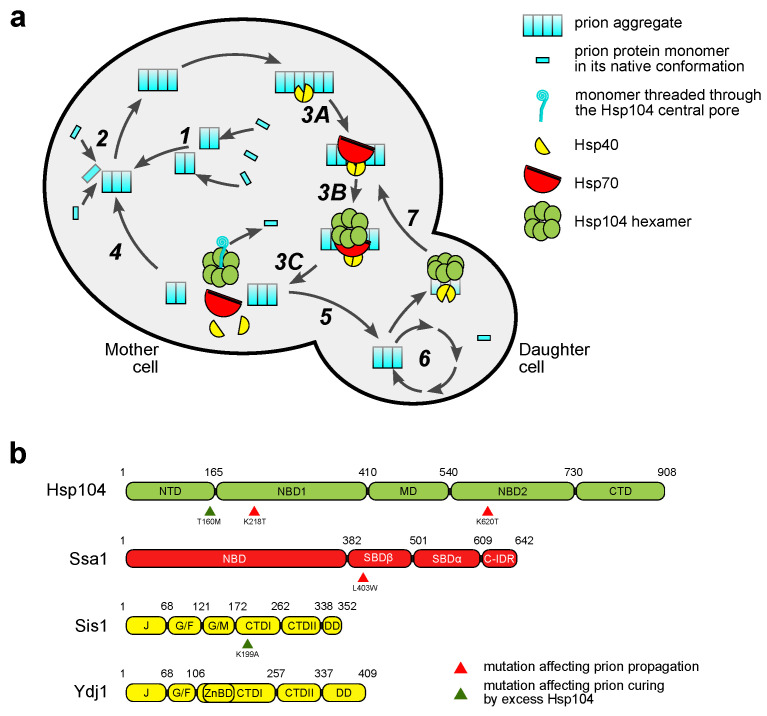
Control of the yeast prion life cycle by PQC machinery. (**a**) A schematic representation of the yeast prion life cycle (based on [41] with modifications). 1—de novo aggregate formation; 2—growth of a newly formed aggregate; 3—chaperone-dependent fragmentation, which includes recognition by Hsp40s, followed by recruitment of Hsp70 (3A), interaction with Hsp104 (3B) and aggregate shearing (3C); 4—continuation of the cycle in mother cell; 5—transmission of prion seed into daughter cell; 6—continuation of the cycle in daughter cell; 7—retaining of an aggregate in mother cell by retrograde transport or asymmetric inheritance. (**b**) The domain structure of the PQC components involved in yeast prion propagation. NTD—N-terminal domain; NBD—nucleotide-binding domain; MD—middle domain; CTD—C-terminal domain; SBD—substrate-binding domain; C-IDR—C-terminal intrinsically disordered region; J—J-domain; G/F—glycine/phenylalanine-rich region; G/M—glycine/methionine-rich region; ZnBD—Zn^2+^-binding domain or region; DD—dimerization domain. Domain boundaries are taken from [42] for Hsp104, [43] for Ssa1, [44] for Sis1, [45] for Ydj1. Arrowheads point to locations of most notable mutations affecting prion propagation or prion curing by excess Hsp104.

**Figure 3 jof-08-00122-f003:**
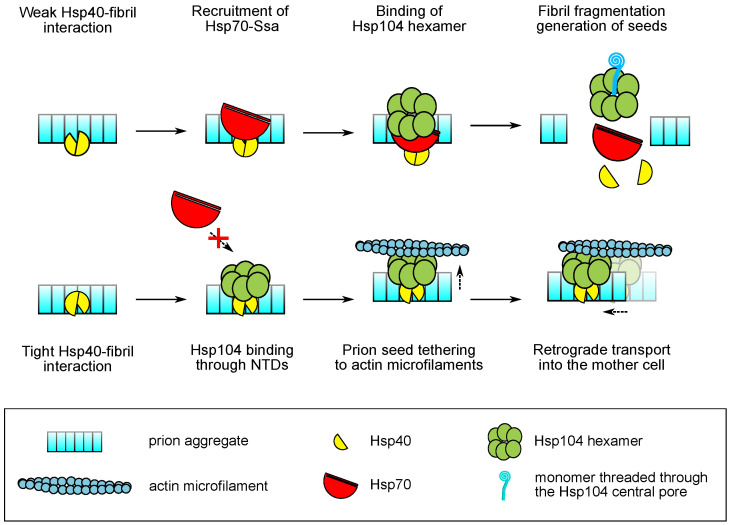
Mechanistic model of the interaction of chaperone proteins and amyloid fibrils. Dashed arrows represent movements of the molecules and formation of complexes. Solid arrows represent the directions of the processes. Crossed arrow indicates an impaired interaction.

**Figure 4 jof-08-00122-f004:**
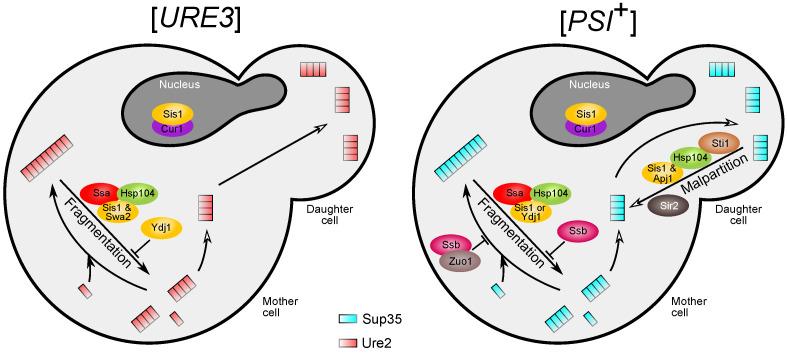
An extended model of yeast prion propagation and its dependence on PQC component activity (modified from [62]). Arrows represent stages of yeast prion life cycle. Empty arrowheads indicate processes related to prion transmission. Arrows with vertical line endings indicate inhibition of a process by a protein or protein complex.

**Table 1 jof-08-00122-t001:** Summary of the effects of PQC proteins on yeast prion propagation.

Protein	Perturbation	Variant	[*PSI*^+^]	[*URE3*]	[*PIN*^+^]	Comment	References
**Hsp104**							
Hsp104	OE	FL	↓↓	↓/none	none		[49,62,63]
		KT *	↓↓	↓↓	↓↓		[49,50]
		ΔN	↑	n.a.	n.a.	Increases [*PSI*^+^] induction rates	[42,65]
	Δ		↓↓	↓↓	↓↓		[49,51,103]
**Hsp70**							
Ssa1	OE	FL	↑/none	↓/none	none	Prevents *HSP104* OE curing of [*PSI*^+^]. Effects on [*URE3-1*] curing are not reproducible, the effect might be strain-specific	[62,73,78,104,105]
		*SSA1-21*	↓↓/none	none	none	Dominant [*PSI*^+^] curing is strain-specific. The effect is more visible in the absence of either *SSA1* or *SSA2*	[63,74,75,105],^†^
Ssa2	OE	FL	↑/none	↓/none	n.a.	Prevents *HSP104* OE curing	[62,79,104]
	Substitution	*SSA2-21*	↓↓	↓↓	n.a.		[63]
Ssa3	OE	FL	↑/none	none	n.a.	Prevents *HSP104* OE curing	[62,79]
Ssa4	OE	FL	↑/↓	none	n.a.	Prevents *HSP104* OE curing	[62,79]
Ssb1/2	OE	FL	↓	none	n.a.	Enhances *HSP104* OE curing	[62,77,78]
	Δ		↑	n.a.	n.a.	Increases [*PSI*^+^] induction rates	[77,80]
**Hsp40**							
Sis1	OE	FL	↓/none	none	none	Enhances *HSP104* OE curing	[100,106,107]
	Depletion	FL	↑/↓	↓↓	↓↓	Effect on [*PSI*^+^] depends on depletion method (Tet-Off or relocalization)	[86,87,100,108]
	Substitution	ΔDD	↓/none	↓	none	Prevents *HSP104* OE curing	[44,89],^†^
		ΔCTD	↓	↓↓	↓↓	Prevents *HSP104* OE curing	[44,89]
		JGF	↓	↓↓	↓↓	Prevents *HSP104* OE curing	[44,89]
		K199A	↓	↓	n.a.	Prevents *HSP104* OE curing	[44,89]
Ydj1	OE	FL	↑/↓/none	↓↓	none	Strain-dependent effect on [*PSI*^+^]	[51,78,100,109]
	Δ		none	none	none		[75,86,110]
Apj1	OE	FL	none	none	n.a.	Enhances *HSP104* OE curing	[94,106,111]
	Δ		none	none	none		[86,110]
Zuo1	Δ		↑	none	none	Increases [*PSI*^+^] induction rates and rates of [*PSI*^+^] curing by mild heat shock	[40,80,86,110,112]
Swa2	Δ		none	↓	none		[86,110]
**Protein-sorting factors**							
Cur1	OE	FL	↑/↓	↓	none	Strain-dependent effect on [*PSI*^+^]	[98,100,109]
	OE	Δ3-22	↑↑	↓↓	n.a.		[62,100]
	Δ		↓/none	↑	n.a.	Increases [*URE3*] induction rates; slightly weakens strong [*PSI*^+^] phenotype and enhances weak [*PSI*^+^] curing by mild heat shock	[98,100,113]
Btn2	OE	FL	none	↓↓	none		[98,100,113]
	Δ		↓	↑	n.a.	Increases [*URE3*] induction rates; enhances weak [*PSI*^+^] curing by mild heat shock	[98,100,113]
Hsp42	OE	FL	↑/↓/none	↓↓	n.a.		[100,113,114]

OE—overexpression, Δ—gene deletion or disruption, FL—full length protein. *—*HSP104KT*-a mutant allele of *HSP104* containing two amino acid substitutions (K218T and K620T) (Figure 1b). Effects of chaperones on prions: ↑—subtle increase in strength of prion phenotype; ↑↑—substantial increase in strength of the prion phenotype; ↓—modest destabilization or weakening of prion phenotype; ↓↓—complete or near complete prion elimination. ^†^—novel results presented in this work (Figure 2).

## Data Availability

Not applicable.

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
