# Peer review of "Differential Interactions of Molecular Chaperones and Yeast Prions"

_jof, 2022, doi:10.3390/jof8020122_

Round 1

Reviewer 1 Report

This is an excellent and thorough review which also includes a small bit of new data and some interesting new analysis. The review was very well-written and quite thoughtful. I found only a single typographical error and have just a few minor suggestions.

Line 118: the authors cite the finding the some variants of [URE3] are "cured" by Hsp104 OE. If my recollection is correct, the data being referred to showed that some variants of [URE3] were destabilized (greater numbers of [ure-o] cells arising) but I don't recall seeing compelling data that any variant was actually cured, meaning eliminated from the cell population. If this is correct, then I think the term "destabilized" should be used here, as I believe it is still true that on [PSI+] is cured by Hsp104 OE. Apologies if I am mistaken and the data is in fact compelling in the opposite direction. I just ask that authors to look again at this distinction since they clearly differentiate elsewhere between phenotypic weakening, prion destabilization, and prion curing.

Line 216: "yeast prion;" should be "yeast prions;"

Lines 223&224: Since this review frequently points out when subtopics have been recently reviewed, then it might be worth mentioning somewhere around this point that this subtopic was recently reviewed in Ref. 88

Figure 2 position: The figure appears several paragraphs after it was mentioned. It would be better if it were moved up to immediately following it's in-text call-out.

Line 246: Consider rephrasing. The situation with Sis1 is a complex point to make. The phrase "most amyloid yeast prions" is awkward because 4 our of approximately 10 is not most. More importantly it can be read to imply that it is not required for some. The reality is that it has been found to be require for all which have been tested, or, for all those for which there is data. I would argue, and I would guess the authors would agree, that it is likely required for all.

Line 306: reference to RAC and rates of [PSI+] induction should also cite:
Amor AJ, Castanzo DT, Delany SP, Selechnik DM, van Ooy A, Cameron DM. The ribosome-associated complex antagonizes prion formation in yeast. Prion. 2015;9(2):144-64. doi: 10.1080/19336896.2015.1022022. PMID: 25739058; PMCID: PMC4601405.

Author Response

Reviewer #1

This is an excellent and thorough review which also includes a small bit of new data and some interesting new analysis. The review was very well-written and quite thoughtful. I found only a single typographical error and have just a few minor suggestions

Authors: We thank the Reviewer for such a positive assessment of our work. All suggestions made by the Reviewer have now been addressed.

Line 118: the authors cite the finding the some variants of [URE3] are "cured" by Hsp104 OE. If my recollection is correct, the data being referred to showed that some variants of [URE3] were destabilized (greater numbers of [ure-o] cells arising) but I don't recall seeing compelling data that any variant was actually cured, meaning eliminated from the cell population. If this is correct, then I think the term "destabilized" should be used here, as I believe it is still true that on [PSI+] is cured by Hsp104 OE. Apologies if I am mistaken and the data is in fact compelling in the opposite direction. I just ask that authors to look again at this distinction since they clearly differentiate elsewhere between phenotypic weakening, prion destabilization, and prion curing.

Authors: The Reviewer is correct, the complete elimination of [URE3] from the population upon overproduction of Hsp104 has not been demonstrated. We corrected the wording accordingly (lines 119-120).

Line 216: "yeast prion;" should be "yeast prions;"

Aithors: The issue was corrected.

Lines 223&224: Since this review frequently points out when subtopics have been recently reviewed, then it might be worth mentioning somewhere around this point that this subtopic was recently reviewed in Ref. 88

Authors: The reference was added to the corresponding sentence (p. 4, line 229).

Figure 2 position: The figure appears several paragraphs after it was mentioned. It would be better if it were moved up to immediately following it's in-text call-out.

Authors: The figure was moved to appear earlier in the text.

Line 246: Consider rephrasing. The situation with Sis1 is a complex point to make. The phrase "most amyloid yeast prions" is awkward because 4 our of approximately 10 is not most. More importantly it can be read to imply that it is not required for some. The reality is that it has been found to be require for all which have been tested, or, for all those for which there is data. I would argue, and I would guess the authors would agree, that it is likely required for all.

Authors: We agree with the Reviewer. The wording was adjusted accordingly (lines 251-252).

Line 306: reference to RAC and rates of [PSI+] induction should also cite:
Amor AJ, Castanzo DT, Delany SP, Selechnik DM, van Ooy A, Cameron DM. The ribosome-associated complex antagonizes prion formation in yeast. Prion. 2015;9(2):144-64. doi: 10.1080/19336896.2015.1022022. PMID: 25739058; PMCID: PMC4601405.

Authors: The reference mentioned by the Reviewer was added to this section (p. 9, lines 311-312).

Reviewer 2 Report

The manuscript by Barbitoff et al. aims to describe the interplay between yeast prions and protein quality control (PQC) machinery and to give a thorough overview of the main molecular chaperones that take part in the prions life cycle and their propagation. Moreover, the authors give detailed information about the main chaperones and their specific effects on different prions as well as new indications for asymmetric segregation of prion seeds. This data can help understand the complex interactions between the PQC machinery and misfolded proteins. Overall, I find the manuscript well-written and important. Nevertheless, some important points should be addressed.

# Page 1, line 15: The keywords do not represent in my opinion the main topics in the paper. “Protein quality control” should be added as well as “prion propagation”.

# Page 1, lines 37-38:  The sentence is misleading since it claims that prions can also have sometimes a beneficial effect on yeast. Most of the microbial prions are not harmful to the cell and are known for many years for their role as information carriers at different conditions.

# Page 2, lines 39-40: The sentences should be rephrased.

# Page 2, lines 72-75: The sentence is too long and should be rephrased.

# Page 3, line 92-93: The authors do not refer to non-amyloid prions, such as [SMAGU+], [ESI+], [GAR+], whose propagation as opposed to what is written in the paper is not Hsp104-dependent, but rather depend on other chaperones, such as Hsp90 and Hsp70. This topic is extensively reviewed in Levkovich, Shon A., et al. "Microbial Prions: Dawn of a New Era." Trends in Biochemical Sciences (2021).

# Page 4, line 144. SSA  is written only in abbreviation and the full name is not mentioned -  Stress Seventy sub-family A. I would include it here. Also, it can be added that SSA represents the major cytosolic Hsp70 family in yeast instead of mentioning that SSA1 is the main chaperones in one place (Page 5, line 151) or SSA2 in a different place  (Page 7, line 241). The abbreviation of SSB can also be mentioned -  Single-stranded DNA binding proteins.

# Page 6, Table 1: Maybe to make it clearer for the reader the Hsp70 and Hsp40 families can be mentioned in the table concerning the members of each family.

# Page 6, line 204-338:  The authors should add subtitles according to the chaperones or the prions. It will help the reader to follow all the data.

# Page 8, Figure 2: It is not clear what is ¼ YEPD as well as [PSI+]s

This figure contains very preliminary data and I don’t think any statements can be concluded from it. I will exclude the figure from the text and take out the relevant information from the Table. In my opinion, the data in this figure don’t contribute a lot to the main scope of the review.   

#Page 9, line 339: This section is really hard to follow as it is now. It is better to subdivide according to the different explanations that are given.

# Page 11, Figure 3: The figure is not clear enough. Especially the arrows that explain the movement of the actin microfilament. It is also very strange that only one alternative model is being addressed in the figure. Also, the terms actin microfilament and actin filaments are not addressed at all in the text.

Minor comments

# Page 2, line 48: “more than 10 yeast prions”. For consistency, the authors should use the word ten instead of 10, the same as written in the abstract (page 1, line 4).

# Page 2, line 85: Hsp100 should be changed to Hsp104.

# Page 4, line 134: The authors should direct the reader to figure1b that where the location of the mutation that was mentioned is shown (T160M0).

# Page 6, Table 1: Increases [URE3] induction rates; enhances weak [PSI+] curing by mild heat shock.”  The dot at the end of the sentence is unnecessary.

# Page 6, Table 1: The table is hard to read. Need more space between the lines.

# Page 12, line 451. ”cell, However”. Should be changed from a comma to a dot.

#Page 10, line 351: “Juxtanuclear Protein Quality Control” should be corrected to “JUxta Nuclear Quality control compartment.

#Page 10, line 351: “Intranuclear Protein Quality Control compartment” should be corrected to “Intranuclear Quality Control compartment”.

Author Response

Reviewer #2

The manuscript by Barbitoff et al. aims to describe the interplay between yeast prions and protein quality control (PQC) machinery and to give a thorough overview of the main molecular chaperones that take part in the prions life cycle and their propagation. Moreover, the authors give detailed information about the main chaperones and their specific effects on different prions as well as new indications for asymmetric segregation of prion seeds. This data can help understand the complex interactions between the PQC machinery and misfolded proteins. Overall, I find the manuscript well-written and important. Nevertheless, some important points should be addressed.

Authors: We thank the Reviewer for the positive assessment of our work. All of the Reviewer’s comments have now been addressed.

# Page 1, line 15: The keywords do not represent in my opinion the main topics in the paper. “Protein quality control” should be added as well as “prion propagation”.

Authors: The keywords were revised accordingly.

# Page 1, lines 37-38:  The sentence is misleading since it claims that prions can also have sometimes a beneficial effect on yeast. Most of the microbial prions are not harmful to the cell and are known for many years for their role as information carriers at different conditions.

Authors: We agree with the Reviewer. The sentence was rephrased.

# Page 2, lines 39-40: The sentences should be rephrased.

Authors: The sentence was rephrased.

# Page 2, lines 72-75: The sentence is too long and should be rephrased.

Authors: The sentence was split and rephrased.

# Page 3, line 92-93: The authors do not refer to non-amyloid prions, such as [SMAGU+], [ESI+], [GAR+], whose propagation as opposed to what is written in the paper is not Hsp104-dependent, but rather depend on other chaperones, such as Hsp90 and Hsp70. This topic is extensively reviewed in Levkovich, Shon A., et al. "Microbial Prions: Dawn of a New Era." Trends in Biochemical Sciences (2021).

Authors: We thank the Reviewer for this suggestion. Non-amyloid prions are now mentioned in the section describing the role of Hsp104. The sentence regarding the requirement for Hsp104 was changed to “Hsp104 is strictly required for maintenance of all known amyloid yeast prions (propagation of non-amyloid prions depends on other chaperone groups - Hsp70 (e.g., for [SMAUG+], [GAR+]) or Hsp90 (e.g., for [ESI+])).”. The corresponding reference has been added (p. 3, lines 93-95).

# Page 4, line 144. SSA  is written only in abbreviation and the full name is not mentioned -  Stress Seventy sub-family A. I would include it here. Also, it can be added that SSA represents the major cytosolic Hsp70 family in yeast instead of mentioning that SSA1 is the main chaperones in one place (Page 5, line 151) or SSA2 in a different place  (Page 7, line 241). The abbreviation of SSB can also be mentioned -  Single-stranded DNA binding proteins.

Authors: The full names of both SSA and SSB Hsp70 subfamilies were added (p. 4, line 147). A misleading statement about Ssa1 was removed.

# Page 6, Table 1: Maybe to make it clearer for the reader the Hsp70 and Hsp40 families can be mentioned in the table concerning the members of each family.

Authors: The information about the chaperone families has been added to the Table. The Table formatting was changed to enhance the readability.

# Page 6, line 204-338:  The authors should add subtitles according to the chaperones or the prions. It will help the reader to follow all the data.

Authors: The subsections were added.

# Page 8, Figure 2: It is not clear what is ¼ YEPD as well as [PSI+]s

This figure contains very preliminary data and I don’t think any statements can be concluded from it. I will exclude the figure from the text and take out the relevant information from the Table. In my opinion, the data in this figure don’t contribute a lot to the main scope of the review.   

Authors: We thank the Reviewer for this suggestion. However, we would like to argue that the data presented are not preliminary and have been reproduced in several independent experiments. To make this point more clear, we now include the results of quantitative analysis of the prion loss rate for both panels. The figure legend was also expanded to provide additional details about the experiments. We believe that these results provide further evidence of the variant specificity of the effects molecular chaperones exert on prions. 

The definition of the 1/4YEPD medium is given in the Figure 2 legend. The [PSI+]S notation used in the previous version of the Figure was used to denote the strong variant of [PSI+]. We removed this notation to avoid further confusion.

#Page 9, line 339: This section is really hard to follow as it is now. It is better to subdivide according to the different explanations that are given.

Authors: We thank the Reviewer for this comment. The section was subdivided.

# Page 11, Figure 3: The figure is not clear enough. Especially the arrows that explain the movement of the actin microfilament. It is also very strange that only one alternative model is being addressed in the figure. Also, the terms actin microfilament and actin filaments are not addressed at all in the text.

Authors: We thank the Reviewer for this suggestion. As the two models of Hsp40/Hsp104 interaction in malpartition are not mutually exclusive, it is hard to depict the two models separately. Hence, we revised the Figure and the corresponding paragraph in the text to better reflect the issue. The movements of molecular complexes were depicted more clearly; definitions of actin filaments were appended to the text.

Minor comments

# Page 2, line 48: “more than 10 yeast prions”. For consistency, the authors should use the word ten instead of 10, the same as written in the abstract (page 1, line 4).

# Page 2, line 85: Hsp100 should be changed to Hsp104.

# Page 4, line 134: The authors should direct the reader to figure1b that where the location of the mutation that was mentioned is shown (T160M0).

# Page 6, Table 1: Increases [URE3] induction rates; enhances weak [PSI+] curing by mild heat shock.”  The dot at the end of the sentence is unnecessary.

# Page 6, Table 1: The table is hard to read. Need more space between the lines.

# Page 12, line 451. ”cell, However”. Should be changed from a comma to a dot.

#Page 10, line 351: “Juxtanuclear Protein Quality Control” should be corrected to “JUxta Nuclear Quality control compartment.

#Page 10, line 351: “Intranuclear Protein Quality Control compartment” should be corrected to “Intranuclear Quality Control compartment”.

Authors: All minor comments have been addressed.

Round 2

Reviewer 2 Report

The manuscript has been sufficiently improved and all of my comments were addressed.